# Understanding and tailoring ligand interactions in the self-assembly of branched colloidal nanocrystals into planar superlattices

Andrea Castelli[1,2], Joost de Graaf[3], Sergio Marras[1], Rosaria Brescia[1], Luca Goldoni[1], Liberato Manna[1] & Milena P. Arciniegas[1]

Colloidal nanocrystals can self-assemble into highly ordered superlattices. Recent studies have focused on changing their morphology by tuning the nanocrystal interactions via ligand-based surface modification for simple particle shapes. Here we demonstrate that this principle is transferable to and even enriched in the case of a class of branched nanocrystals made of a CdSe core and eight CdS pods, so-called octapods. Through careful experimental analysis, we show that the octapods have a heterogeneous ligand distribution, resembling a cone wrapping the individual pods. This induces location-specific interactions that, combined with variation of the pod aspect ratio and ligands, lead to a wide range of planar superlattices assembled at an air–liquid interface. We capture these findings using a simple simulation model, which reveals the necessity of including ligand-based interactions to achieve these superlattices. Our work evidences the sensitivity that ligands offer for the self-assembly of branched nanocrystals, thus opening new routes for metamaterial creation.

---

[1] Istituto Italiano di Tecnologia, Via Morego 30, 16163 Genova, Italy. [2] Dipartimento di Chimica e Chimica Industriale, Università degli Studi di Genova, Via Dodecaneso, 31, 16146 Genova, Italy. [3] SUPA, School of Physics and Astronomy, The University of Edinburgh, King's Buildings, Peter Guthrie Tait Road, Edinburgh EH9 3FD, UK. Correspondence and requests for materials should be addressed to J.d.G (email: j.degraaf@ed.ac.uk) or to M.P.A. (email: milena.arciniegas@iit.it)

The self-assembly of colloidal nanocrystals enables the formation of structures possessing a complexity often not attainable via conventional approaches[1–3], and of which the properties can differ strongly from their single building blocks[4,5]. In addition, the inherent thermodynamic tendency of these objects to form ordered structures can make self-assembly a cost-effective means to fabricate metamaterials[5,6]. However, engineering the building blocks to produce a desired superlattice requires a careful tailoring of interactions occurring between nanocrystals at different length scales[4,5,7–10]: (i) the self-assembly's long-range symmetry is commensurate with that of the far-field guiding forces; (ii) short-ranged interactions lead to the fine ordering between neighboring particles. The latter can thus overcome the symmetry established by long-ranged interactions, enabling nanocrystals to self-organize into more exotic superlattices[1,2,11].

Typically, strong repulsive and attractive forces coming from, e.g., the nanocrystal shape[12] and the ligand molecules covering the particle surfaces[13], hinder the formation of ordered structures. However, extensive work on colloidal synthesis and post-synthesis treatments of nanocrystals has enabled the sensitive tuning of interaction strength to thermal energy required to overcome some of these issues[14–18]. This is achieved, for instance, by using suitable ligands to both extend the solubility of the building blocks and facilitate their integration within a polymer matrix[19–23]. Yet, controlling the ligand coating of a nanocrystal is not easy: the density and distribution on the particle surface is typically the result of an equilibrium between the interaction energy of the (various) nanocrystal facets with the solvent and their stabilization by organic molecules, which leads to local ligand variation[24–26].

In the context of ligand-mediated interactions, it has been experimentally shown that mastering the particle shape (and thereby implicitly the local ligand distribution) enables the directed assembly of multi-component structures via shape-complementary interactions[11,27–29]. The same driving force was also recently reported to govern the assembly of complex shapes such as irregular hexagonal nanoplates[25] and tetrahedral nanocrystals[30]. However, tuning the assembly of branched particles by their surface–ligand coverage has remained challenging, due to the multiple ligands used in their synthesis[31–33]; a prerequisite to form the particles' unique shape. These ligands can make the particles interact differently through different facets, which further complicates achieving the careful balance of forces required to control their self-organization.

In this article, we investigate the self-assembly into planar superlattices of nanocrystals with a branched shape that are made of a CdSe core and eight CdS pods, so-called octapods[31], when solvent evaporation forces them to form highly ordered structures at the liquid–air interface. To date, several types of octapod assemblies have been studied and hypotheses have been formulated on their mechanisms, based on microscopy analysis of

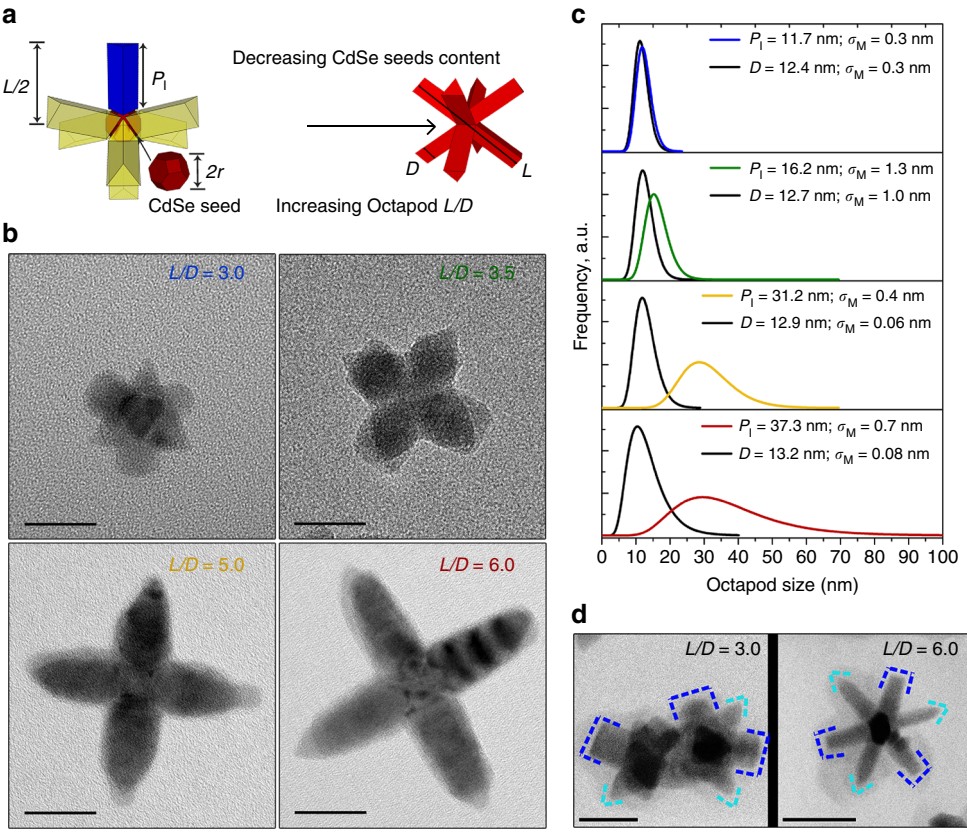

**Fig. 1** Details of the as-synthesized octapods with different pod length. **a** Sketches illustrating the geometrical parameters of an octapod nanocrystal and the variation in octapod aspect ratio. $L$ and $D$ represent the octapod length and diameter, respectively. The tip-to-center length of the pods is represented by $L/2$ and the actual length of the pod is given by $P_\text{I}$. The truncated octahedral CdSe seed is indicated in red and has radius $r_\text{CdSe seed}$. **b** BFTEM images of representative particles for the four octapod aspect ratios $L/D$ studied in this work. Scale bars=20 nm. **c** Volume-weighted size distribution for $D$ and $P_\text{I}$ octapod parameters determined by XRD analysis. Data are presented as the average values and their standard error of the mean, $\sigma_\text{M}$. **d** Inverted-contrast HAADF-STEM views of the octapods evidencing the two families of tips observed in both octapods with small and large $L/D$ aspect ratio. That is, four flat-terminated pods (framed in blue) and four pods with sharp tips (framed in cyan). Scale bars=20 nm and 50 nm for the small and large $L/D$ aspect ratio, respectively

the geometry of the structure formed and by computational modeling of the assembly process[2,29,34–36]. These studies have significantly improved our understanding of the assembly mechanisms of this class of branched nanocrystals. Yet, a detailed analysis of the role of the surface ligand coating of the particles on their self-assembly behavior has so far remained unexplored. In addition, although the synthesis of octapods has been studied before using several ligands and seeds prepared at high temperatures[31,32], the facile tuning of the pod length to achieve octapods with very short pods had not yet been attempted. Here, through a combination of experimental and theoretical/simulation analysis, we demonstrate how the anisotropy, in terms of both particle shape and ligand coverage, dictates the octapods' final configuration. We first tune the synthesis parameters to control the length of the pods and thereby their aspect ratio. Next, we assess the impact of the aspect ratio on the ligand distribution using Fourier transform infrared spectroscopy (FTIR). From our analysis, we hypothesize that a cone-like distribution of ligands on the pods of the particles is responsible for their interlocked configuration. Our simulation model shows that the experimentally observed structures may indeed be recovered, when such an (effective) interaction potential is assumed. We further show experimentally that the modulation of the octapod–octapod interaction by facile ligand-exchange strategies results in a variety of assembled geometries. The obtained structures range from square lattices to arrays of closely packed interlocking chains. These structures are also predicted using our simple simulation model, which displays similarly strong sensitivity to the tip and shaft interactions. This implies that ligand-modulated interactions are key to explain the experimentally observed structures and with this, we complete the picture of the process by which octapods self-assemble.

## Results

**Synthesis**. Changes in the octapod size were possible by controlling the amount of injected $Cu_{2-x}Se$ seeds, from ca. 2 nmol to 0.2 nmol, while keeping constant the amount of the other precursors and ligands in their synthesis. Briefly, a mixture of CdO, CdCl2, TOPO (tri-n-octylphosphine oxide), TOP (tri-n-octylphosphine), ODPA (octadecylphosphonic acid), and HPA (hexylphosphonic acid) was heated up to 350 °C under $N_2$ flow. Next, a solution containing the pre-synthesized seeds, plus an aliquot of S and additional TOP, were quickly injected at this temperature. The resulting mixture was cooled down to room temperature after 10 min. The obtained suspension of octapods in toluene was purified to remove the excess of organics that derived from the synthesis. Full details of the protocol are provided in Supplementary Note 1 and Supplementary Figs. 1 and 2. We selected four octapod batches, with different aspect ratio, based on the quality of their shape and their ability to form stable suspensions. Representations of the octapods are shown in Fig. 1a, which also indicates the relevant octapod dimensions: tip-to-tip pod lengths (L), pod diameter (D), and actual pod length ($P_I$), that is defined as $P_I = L/2 - r_{CdSe\ seed}$ with $r_{CdSe\ seed} = 12.9 \pm 1.4$ nm, that represents the mean ± standard deviation, in this case evaluated from the projection of 150 nanocrystals in transmission electron micrographs (TEM). The sketches further show the tip and the lateral facets of the pods, henceforth labeled as "tips" and "shafts", respectively. Figure 1b displays a collection of high-magnification bright field TEM (BFTEM) of single particles evidencing the variation in octapod sizes, with lengths L ranging from ~40 to 90 nm, for pods with diameters D of ~13 nm, as calculated from TEM analysis (see Supplementary Table 1 and Supplementary Fig. 3). The collected X-ray diffraction (XRD) patterns and photoluminescence spectra of the selected octapods are shown in

Supplementary Fig. 4. Further analysis of the XRD patterns (see Methods for details) confirmed the obtained average values of $P_I$ and D of the as-synthesized octapods, see Fig. 1c.

We also observed that the variation in the octapod aspect ratio does not affect the pod cross section, since that depends on the basal crystalline plane, as demonstrated in the single tilt series of the selected octapods by high-angle annular dark field scanning TEM (HAADF-STEM), see Supplementary Fig. 5 and volume reconstructions of the particles in Supplementary Movie 1 (octapods with L/D = 3.0) and Supplementary Movie 2 (octapods with L/D = 6.0). As can be appreciated from Fig. 1d, the octapods' two families of pods—four with a flat tip and four with a sharp tip —are also observed in smaller structures, as we reported previously for larger octapods[37]. Crucial to the following is that our synthesis involves four phosphorus-based ligands (used at the same concentration for all the octapod sizes): HPA, ODPA, TOP, and TOPO, see Supplementary Table 2.

**Assembly of octapods and their ligand distribution**. To study the behavior of the selected batches of octapods, when stabilized with their native ligands, we drop cast an aliquot of each suspension in hexane on the top of a diethylene glycol layer used as a substrate[27] to allow for planar superlattices to form, see details in the Methods and Supplementary Fig. 6. The absorbance of octapods suspensions as a function of the different concentrations studied in their assemblies is shown in Supplementary Fig. 7. Figure 2 shows a collection of HRSEM images of the planar superlattices formed by the selected octapods at a fixed concentration of 5 nM in hexane; low-magnification SEM images are provided in Supplementary Figs. 8 and 9. For particles with a small aspect ratio ($L/D \leq 3.5$), the superlattices are formed by octapods standing on four pods. In the case of the assembly from particles with L/D of 3.0, the octapods are less densely packed and they are preferentially in contact with neighboring ones through their tips, forming short-range ordered domains. Fast Fourier transformation (FFT) analysis of collected large area HAADF-

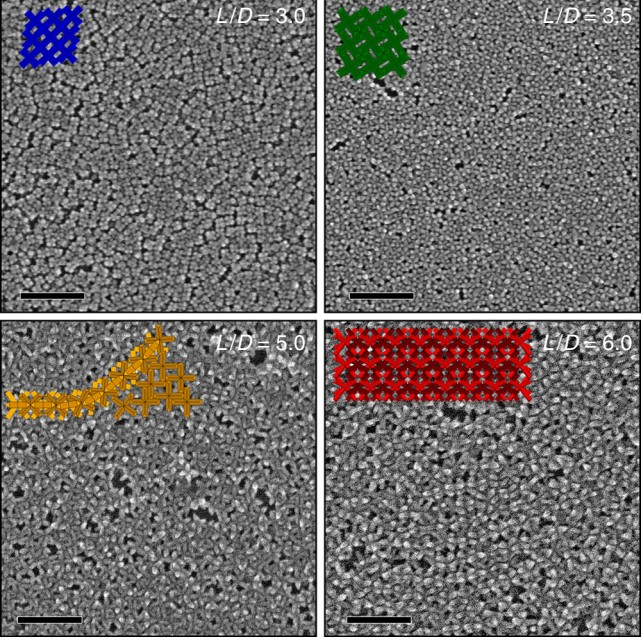

**Fig. 2** HRSEM images of self-assembled planar superlattices of octapods with different aspect ratio. Octapods with a small L/D stand on four pods with domains of square lattices, while high L/D particles exhibit a packing of interlocked chains. Scale bars=200 nm. The embedded sketches highlight the observed configurations of octapods for each aspect ratio

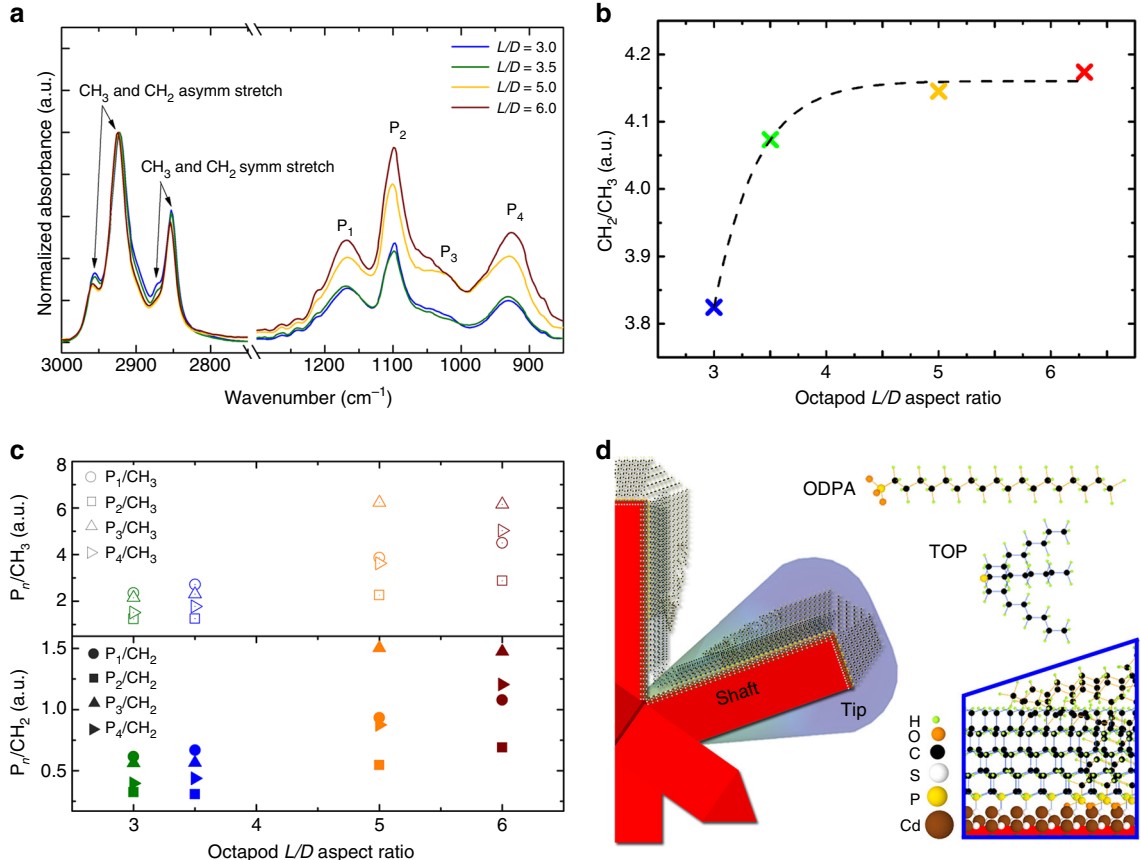

**Fig. 3** Surface characterization of octapods with different aspect ratio. **a** FTIR spectra collected from the selected octapods; $P_{n=1...4}$ indicates the peaks related to P–O(H) and P = O stretching modes of the ligands, mainly the P = O moiety from TOPO and $PO_3^{2-}$ from phosphonic acids. **b** $CH_2/CH_3$ asymmetric stretching peak ratio for different $L/D$ proving an increase in the amount of long-tailed ODPA on longer shafts, until the signal covers that of the short ligands, coming from the shaft portion near the core. **c** $P_n/CH_2$ and $P_n/CH_3$ ratios for the studied octapods. Both plots show an increase of the $P_{n=1...4}$ compared to the $CH_2$ and $CH_3$ asymmetric stretching modes. **d** Sketch of an octapod pod showing a cone-like ligand distribution predicted from FTIR analysis: short ligand molecules (TOP) wrap the regions near the core, while long ones (ODPA) are preferentially bound toward the tips when increasing octapod aspect ratio. HPA is also present on the shafts of the pods, but there is no evidence of preferential attachment

STEM images of the octapod superlattice shows that the particle periodicity remains constant within domains (ca. 25.4 nm, see Supplementary Fig. 10). A similar analysis conducted on smaller areas formed by 10–30 particles, shows that there are domains containing octapods with a 90°-orientation with respect to closer particles, forming a square lattice configuration (Supplementary Fig. 11). For octapods with $L/D = 3.5$, we observe more compact superlattices and octapods appear in full contact through their shaft. By further increasing the pod length of the particles, superlattices are formed by fully-interlocked chains of octapods with domains containing up to five aligned linear structures that reach 1 μm in length. The transition from square lattices to interlocked chains occurs at $L/D = 5.0$, when short chains are already observed along with domains formed by tilted octapods. Finally, the observed self-assembled configurations of octapods are not affected by varying the octapod density in the starting solution (Supplementary Figs. 12 and 13) when working with particles of $L/D < 5.0$.

In contrast, octapods with higher aspect ratio exhibited different assembled configurations depending on the particle concentration; at high particle concentration, there is a strong aggregation of octapods. The ability of octapods to form interlocked structures in 2D superlattices has been previously documented by our group[2,29] for nanocrystals with $L/D$ of 7.0 and in 3D superstructures for octapods with $L/D$ of 8. It becomes clear that the origin of such behavior strongly depends on the

octapod's geometrical parameters. The tight control over pod-length achieved through our current synthesis of octapods is a pre-requisite for shedding light on the contribution of ligands on their complex ordering.

To elucidate the role of ligands in the assembly, we investigated the octapods' surface for the selected batches via FTIR. We also approached the analysis via High-Resolution Nuclear Magnetic Resonance (HR-NMR) using phosphorus-31 NMR signals. However, due to the sensitivity of the technique and primarily the lack of well-dispersed suspension of octapods at the (high) concentrations required for the analysis, a complete study of the chemical composition of octapod ligand shell was significantly more limited, see Supplementary Note 9. Thus, we focused our attention on a detailed ATR-FTIR analysis. To avoid changes on the footprints of the bound ligands on the spectra, due to preferential orientations of the particles, we performed the studies on dried samples of octapods prepared via drop casting followed by a fast solvent evaporation at open air. Collected FTIR spectra for particles stabilized with their native ligands are shown in Fig. 3a. We observe that the surfaces of the selected octapods have similar features, which is reasonable as the particles were synthesized with the same type and amount of phosphorus-based ligands. These features are found in the region between 1250 and 950 cm$^{-1}$ that collects signals from bound ligands with P–O moieties ($P_{n=1...4}$ peaks in Fig. 3a related to TOPO, HPA, and ODPA) and the region between 3000 and 2800 cm$^{-1}$ that

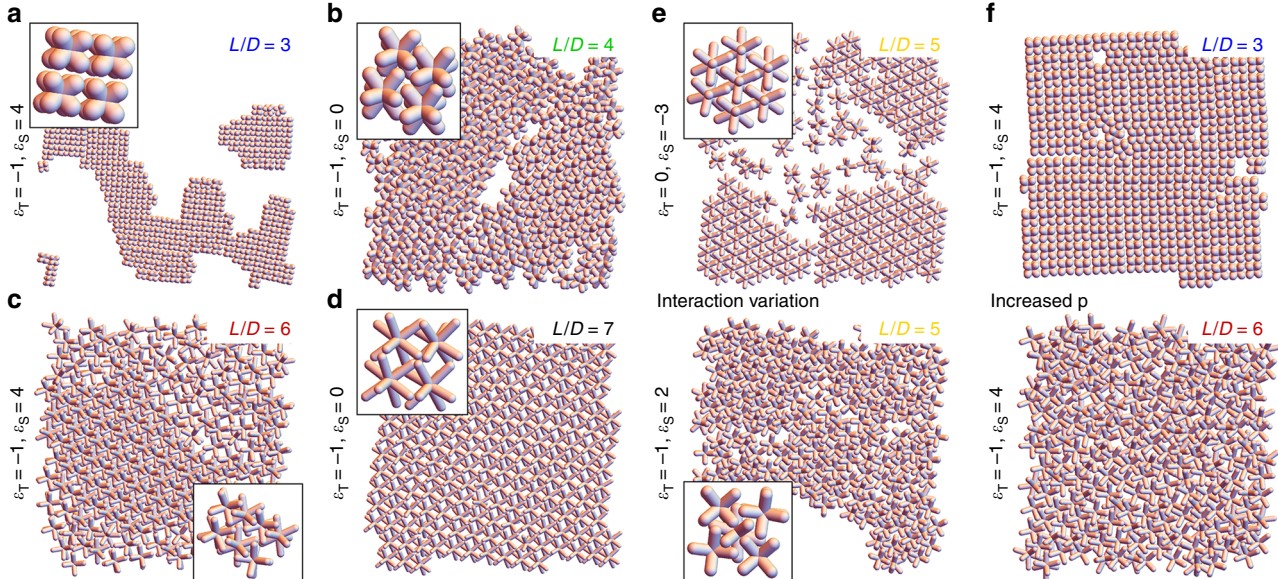

**Fig. 4** Monte Carlo Simulations of 2D superlattices of octapods with different pod length. **a–d** Snapshots showing a typical top-view configuration of the octapods at the end of our simulation run for different $L/D$. The value of the shaft interaction parameter $\varepsilon_S$ and the tip interaction parameter $\varepsilon_T$ are provided to the left of each snapshot, both are given in $k_BT$. A description of the crystal structure is given in the main text; the insets show a zoom-in to better visualize the structure. **e** Variation of the parameters $\varepsilon_S$ and $\varepsilon_T$ for octapods with $L/D = 5.0$ tune the octapod arrangement from ballerina (for $\varepsilon_S = -3$ and $\varepsilon_T = 0$) to interlocking (for $\varepsilon_S = 2$ and $\varepsilon_T = -1$). **f** The pressure was increased with respect to (**a**) and (**c**)

shows the $CH_2$ and $CH_3$ stretching modes deriving from the alkyl chain(s) of the bound phosphorus-based ligands; see details in Supplementary Note 5.

We find, however, the following division of signals: (i) octapods with enhanced $P_{n=1...4}$ peaks ($L/D \geq 5.0$) and (ii) octapods with weaker signals in the same region, but with relatively more intense $CH_3$ stretching modes in the region of high wavenumbers ($L/D \leq 3.5$). This observation indicates that there is a difference in the surface ligand distribution between octapods with small and large $L/D$. This division corresponds to the one found in the assemblies of octapods, suggesting that ligand-mediated interactions play a key role in defining the morphology of the observed geometries.

We assess the location of the different ligands on the octapod surfaces (pod's shafts and tips) by first performing a comparative analysis between the deconvoluted areas of the $CH_2$ and $CH_3$ asymmetrical peaks. That is, we examine the $CH_2/CH_3$ ratio observed in the FTIR spectra of the octapods, see Fig. 3b. We found that there is an increased signal from the $CH_2$ asymmetrical stretching mode, which saturates for large aspect ratios. The absorptions coming from the termination of the alkyl chains ($CH_3$ vibrational modes) becomes less pronounced when compared to those from the bulk of the chains ($CH_2$ vibrational modes), therefore, the length of the alkyl chain wrapping the pods increases with $L/D$. Thus, on average the surfaces are stabilized by a thicker ligand shell. In other words, they are preferentially covered by ODPA, which is the ligand with the longest alkyl chain used in the synthesis (Supplementary Table 2). For different $L/D$, the pods are structurally identical, hence the increase of ODPA ligand cannot be related to preferential facet stabilization. Instead, the restricted space between pods in the region where they meet the central core and the relatively long alkyl chain of the ODPA molecules, reduce the ODPA's access to pods' surfaces in that region, due to strong steric repulsion forces. Based on this, it is reasonable to assume a thicker ODPA ligand wrapping of the pods toward the tips.

Next, we evaluated the ratio between the deconvoluted area of the $P_{n=1...4}$ with respect to that of the $CH_2$ and $CH_3$

asymmetrical stretching peaks, that is $P_n/CH_2$ and $P_n/CH_3$, for all the studied octapods, Fig. 3c. The observed increase in both estimated ratios indicates that the P-O(H) and P = O stretching mode signals (related to the $P_{n=1...4}$ peaks) become more intense compared to those from the $CH_2$ and $CH_3$ asymmetrical stretching peaks. That is, the amount of P-O groups per alkyl chain in the ligand shell is increasing with the aspect ratio. This observation brings us to the following reasoning: the increased amount of ODPA molecules (1 $PO_3^{2-}$ moiety per 18 carbons) in particles with larger aspect ratio (Fig. 3b) may result in an increase in the amount of P–O moieties only if octapods with low aspect ratio are stabilized by TOP (no P-O moieties and 24 carbons), or alternatively by TOPO (1 P = O per 24 carbons). The presence of TOP- or TOPO-rich shafts would not have an impact on the thickness of the ligand shell, since the alkyl chain of these ligands is identical. However, TOPO-rich shafts are unlikely because it would induce a strong change in the region of the $P_{n=1...4}$ with a marked sink in intensity of the $P_2$ peak. Assuming, instead, a HPA-rich shell (1 $PO_3^{2-}$ moiety per 6 carbons) for octapods with short pods, an increase of ODPA molecules would have forced a reduction in the amount of P-O groups per carbon, and thus, in the intensity of the $P_{n=1...4}$ peaks, which is not observed in the FTIR spectra. Therefore, the most straight forward explanation for the observed absorption behavior is the presence of a TOP-rich shell for short pods. In addition to this, the increase in the $P_2$ peak intensity with increasing $L/D$ observed in Fig. 3c indicates that HPA molecules help to stabilize the shaft of the pods and promote their growth, as demonstrated also by further experiments on the impact of the native ligands in the synthesis of octapods, see Supplementary Figs. 14 and 15.

This closer examination of the FTIR spectra allowed us to formulate an experimental model for the ligand coverage of the pods, as shown in Fig. 3d. For octapods with a large $L/D$, shorter ligands, like TOP, are attached closer to the core and longer ones, like ODPA, which are found preferentially around the tips, giving rise to a cone-like ligand shell. Octapods with a small $L/D$ are mainly stabilized by shorter ligands, i.e., HPA and TOP. The crossover between the two regimes is at $L/D \approx 5$.

**Monte Carlo simulation of the octapod assemblies**. We used Monte Carlo (MC) simulations to understand the experimental observations in our as-synthesized octapod samples, extending previous models for their self-assembly to account for the new insights into the ligand distribution on the octapods. These simulations consisted of a standard isothermal-isobaric (NPT) MC approach in a quasi-2D setup that modeled the localization of the octapods at the interface, see the Methods for the details. Our model is comprised of four intersecting spherocylinders with a hard overlap potential to represent the excluded volume interactions between the octapods, as originally used to study the structures formed by drop-cast octapods on a hard substrate[36]. We endow this model with square-well/shoulder interactions around the tips, with interaction strength $\varepsilon_T$, and similarly around shafts, with interaction strength $\varepsilon_S$, to capture the physics of the surface interactions as well as to obtain the interlocking configuration. To model the heterogeneous ligand distribution, the shaft potential becomes weaker towards the core of the octapod (Supplementary Fig. 16). Positive values of the interaction strength indicate repulsion, while negative values imply attraction, see details in Supplementary Note 6. Here, it should be noted that the effective interactions model the combined effect of ligands and other (unknown) interaction potentials present in our system, such as van der Waals (vdW) attractions. This is the reason for allowing the net interaction to be attractive on certain areas of the octapod and repulsive on others in our parameter analysis, see Supplementary Notes 6 and 7. This modeling proves to be more predictive than our previous attempts that accounted accurately for a single interaction, such as vdW attractions, but did not incorporate all the relevant details of the surface interactions.

We only observed self-assembled structures when either $\varepsilon_S$ or $\varepsilon_T$ is nonzero, implying that some short-ranged attractions or repulsions must be present to form ordered structures, especially the interlocking configuration—at sufficiently high pressures disorganized agglomerates are formed. The primary result of our Monte Carlo simulations is presented in Fig. 4a–e, which shows representative snapshots of some of the structures found by varying $L/D$, $\varepsilon_S$, and $\varepsilon_T$, at a constant pressure $p$; the full parameter variation is provided in Supplementary Figs. 17–21.

Our model produces square-lattice-like packings for $L/D = 3$ with a few line defects (Fig. 4a), as well as slightly skewed variants of the square lattices for $L/D = 4$ (Fig. 4b). Supplementary Figs 17 and 18 reveal that for these aspect ratios, the tip interaction ($\varepsilon_T$) is the only relevant contribution—the shafts are too small to influence the structure. Any difference in the structure between these two aspect ratios is therefore driven purely by the geometry in our model. When we increased the octapod aspect ratio to $L/D = 5$, we found that superlattices with an interlocking configuration started to appear, accompanied by ordered and disordered domains of octapods standing on four pods (Supplementary Fig. 19).

This observation agrees with our experimental findings. Both interlocking and ballerina configurations emerge depending on the choices for the shaft interaction potential (Fig. 4e). Attractive shaft interactions of $-3k_BT$ lead to the ballerina configuration, when the tips do not have a short-ranged interaction, that is for $\varepsilon_S = -3$ and $\varepsilon_T = 0$ in Fig. 4e. Repulsive shaft interactions of $+2k_BT$ and weak tip attractions of $-1k_BT$ give rise, instead, to interlocking configurations when $\varepsilon_S = 2$ and $\varepsilon_T = -1$, Fig. 4e.

Increasing the octapod aspect ratio to $L/D = 6$, we also obtained these structures, but the interlocking configurations are far more clearly defined (Fig. 4c), albeit for slightly different interaction parameters than for $L/D = 5$ (Supplementary Fig. 20). This further evidences, as in the experiment, that the length of the pods factors into obtaining regular self-assemblies, although it is not as critical as the ligand distribution. Strikingly, within the limitations of our model, the interlocking configuration is only obtained using repulsive shaft interactions. This suggests that on the shafts, ligand interactions dominate over the strictly attractive van der Waals interactions. Our findings also cast new light on our previous observations for the ballerina-configuration[27]. We had hypothesized that the formation of such network in a polymer matrix could be due to rotational constraints imposed by

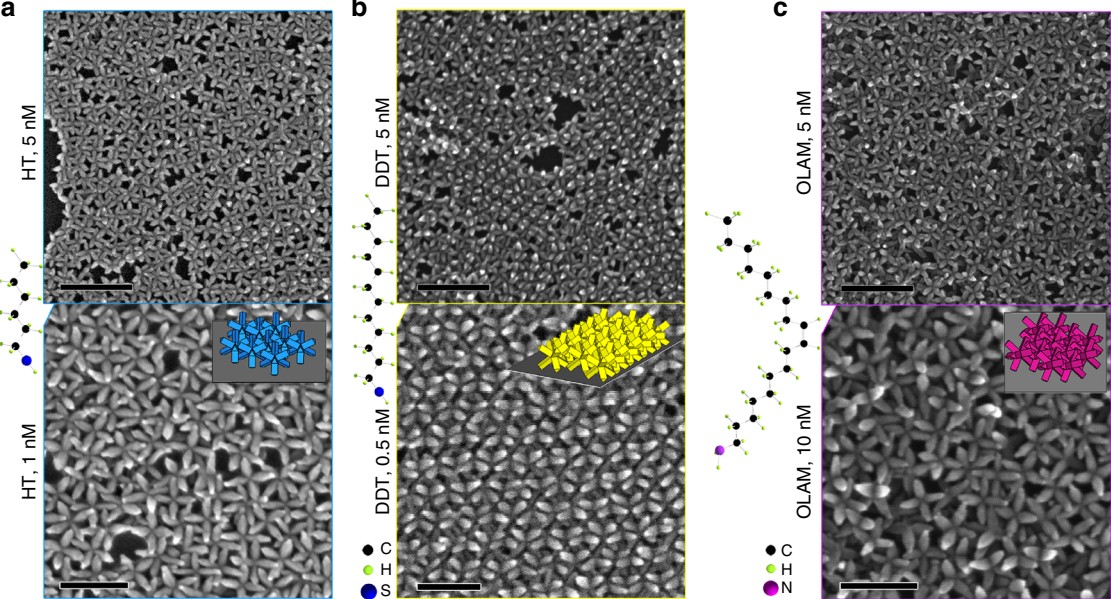

**Fig. 5** Self-assembled 2D superlattices of octapods with $L/D$ of 5.0 stabilized with different ligands. **a** Square lattice of HT-ligand exchanged octapods at 5 nM and 1 nM. **b** Interlocked chains of octapods for DDT-exchanged octapods at 5 nM. A tight packing of chains is observed at lower concentrations. **c** OLAM-exchanged octapods standing on four pods and forming binary square lattice in small domains at 5 nM and even at high concentrations (10 nM). The side representations next to the panels show the atomic structure of the used ligands. The embedded sketches depict the observed octapod configurations. Scales bars: for 5 nM octapod concentration, 200 nm, and for the 1 nM, 0.5 nM, and 10 nM octapod concentration, bottom panels) 100 nm

the organic–air interface. Here, we show that the polymer could also have modulated the interaction between the octapods to lead to the ballerina configuration. For $L/D = 7$ we discovered a new type of regular self-assembly, one where the tips of the octapods touch to form a tip-touching square lattice structure, see Fig. 4d and Supplementary Fig. 21. However, this structure is not observed in experiment, presumably due to it being mechanically unstable against drying forces or lying outside of the $L/D$ range that we considered.

Finally, we verified the stability of our structures by varying the pressure. Typically, our small $L/D$ results are only weakly dependent on the pressure that we chose (top panel in Fig. 4f); Supplementary Figs 22–26 show that the structures that we found for a single pressure are obtained over a much wider range. For $L/D \geq 5.0$, however, we found that sufficiently high pressures induce more random aggregates, rather than interlocked structures (Fig. 4f and Supplementary Fig. 25), presumably because of the self-assembly being driven far out of equilibrium.

**Assembly of ligand-exchanged octapods**. To experimentally confirm the tunability of octapod interactions/assemblies predicted by the simulations, we introduced new molecules on their surface through a post-synthesis ligand exchange strategy (see details in the Methods) and prepared 2D superlattices, following the interfacial self-assembly approach described above. Specifically, we used thiol-based ligands of different length, from 6 to 18 carbon atoms: hexanethiol (HT), dodecanethiol (DDT), and octadecanethiol (ODT) to assess the effect of different ligand shell thickness on the modulation of the shaft–shaft interaction between neighboring pods of the octapods with $L/D$ of 5.0. The theoretical analysis (Fig. 4e) suggests that for this aspect ratio the configuration of octapods can change depending with variation of the shaft and tip interaction. We also performed ligand exchange using each native ligand alone (TOP, TOPO, HPA, and ODPA), as well as oleylamine (OLAM) to evidence the sensitivity of the resulting assembly of octapods to both small and large variations of potential interactions, respectively. Our FTIR spectra indicate that the exchange was effective for the following ligands: HT, DDT, and OLAM (Supplementary Fig. 27). Detailed analysis of the collected spectra suggests a thinner ligand shell present on the surface of the HT-ligand exchanged octapods, compared to that of DDT or OLAM-ligand exchanged particles (Supplementary Fig. 28). Octapod ligand exchanges with HPA and ODPA did not result in stable solutions by following the described protocol, and thus they were not used for the analysis. The observed lack of stability of octapods when using ODPA is attributed to the observed limited solubility of the ligand in toluene at room temperature. In the case of HPA, it denotes its weak binding to the octapod surface.

Figure 5 shows a collection of self-assembled 2D superlattices for octapods stabilized with different molecules, starting from those formed when working with a particle concentration of 5 nM. Note that initially the same as-synthesized octapods presented a mixture of both square lattice and interlocked chains (Fig. 2, octapods with $L/D$ of 5.0).

Remarkably, their arrangements change to (i) solely square lattices (Fig. 5a) for HT ligand-exchanged octapods, (ii) interlocked chains (Fig. 5b) for DDT ligand-exchanged particles, and (iii) octapods standing on four pods with small domains of binary square lattices from OLAM ligand-exchanged particles (Fig. 5c). As can be appreciated from Fig. 5a and c, the presence of either a short molecule as HT or a long one as OLAM on the surface of octapods fully supresses the formation of interlocked chains, independent of the particle concentration (Fig. 5a and c). This is also observed for octapods with an $L/D$ of 6.0 after OLAM

ligand exchange (Supplementary Fig. 29). A square lattice is the dominant configuration of octapods under the aforementioned conditions, which, when combined with the insights from our modeling, indicates that exchange with such molecules does not induce sufficient shaft-to-shaft repulsive interactions between neighboring octapods to favor their interlocking. In the case of OLAM-ligand exchanged octapods, this behavior prevails even at higher particle concentration (10 nM), while interlocked chains appear in the superlattices formed from HT-ligand exchanged above a concentration of 5 nM (Supplementary Fig. 30). A higher level of ordering is also observed from HT-ligand exchanged octapods at concentrations below 5 nM (Fig. 5a, 1 nM), for which regions containing binary square lattices are observed[38]. These observations indicate a stronger modulation of the shaft-to-shaft interactions induced by thiols compared to those induced by the amine, most likely due to the length of the latter type of ligand, which might more effectively screen particle interactions.

Interestingly, by introducing a longer thiol molecule (DDT) on the surface of the octapods we found interlocking chains at 5 nM, that are well-compacted and aligned at a low particle concentration of 0.5 nM, reaching up to 2 μm length (Fig. 5b). In contrast, disordered aggregates of octapods are observed from TOPO-ligand exchanged particles at a concentration of 5 nM (Supplementary Fig. 31). It should be noted that, when working at concentrations above 5 nM, the superlattices from the as-synthesized octapods and the OLAM-ligand exchanged ones, contain a few randomly oriented interlocked chains and octapods standing on four pods, respectively (Supplementary Figs. 12 and 30). The latter also present small domains of binary square lattices. Clearly, the switching between configurations from octapods capped with HT, DDT, or OLAM—not observed with the as-synthesized particles—evidences that introducing new ligand molecules can modulate the interactions between the different parts of the pods.

Finally, we applied the ligand exchange protocol, using the same molecules, to smaller structures (octapods with an $L/D$ of 3.5). Contrary to what was described above for octapods with $L/D$ of 5.0, we did not find differences in the arrangement of octapods with $L/D$ of 3.5 in their superlattices (Supplementary Figs. 31 and 32). We observed domains of square lattices from all the exchanged particles, as formed by the as-synthesized ones. This observation further confirms that the length of the pods in the branched nanocrystals plays a key role to make them highly sensitive to modulations of interaction through ligands, which must predominantly impact the shaft area, in agreement with the simulation model.

## Discussion
Our study demonstrates the power of ligands to directly tune the complex interaction between the different parts of branched nanocrystals, thereby allowing for the modulation of their self-assembled planar superlattices. Specifically, we synthesized octapod-shaped nanocrystals with four different aspect ratios, through variations on the injected seed amount. We hypothesized a cone-like distribution of ligands on the pod surface of octapods through a combined analysis of their 2D self-assemblies and their native surfaces. The preferential binding of long ligand molecules toward the pod tips appears responsible for the remarkable interlocking of octapods. Our simulation modeling and subsequent experiments strongly support that a small difference between the interaction of both part of the pods, tips and shafts, is sufficient to generate a variety of structures from particles with a fixed size by selectively ligand binding. Our study thus advances understanding of fundamental impact of ligand interactions on the ordering of particles with multiple branches and opens a new

route towards exploiting these to generate a wealth of metama-terials difficult to attain by other means.

## Methods

**Synthesis of branched nanocrystals**. Octapods were synthesized using different amounts of preformed $Cu_{2-x}Se$ seeds to tune their pod size. Details are provided in Supplementary Note 1. The provided concentration of octapods (in nM) were calculated based on the Cd contain (in ppm) determined through an elemental analysis conducted on an iCAP 600 Thermo Fisher Inductively Coupled Plasma Optical Emission Spectrometer (ICP-OES). The samples were digested overnight in a $HNO_3$/HCl solution and diluted in deionized water. All the suspensions were filtered (using PTFE filters) before analysis. The collected absorbance of octapod suspensions at different concentrations is provided in the Supplementary Fig. 7.

**Ligand-exchange protocol**. Ligand exchange was carried out with eight different ligands, that is HT, DDT, ODT, HPA, ODPA, TOPO, TOP, and OLAM and using the same protocol to assess the surface responsivity of the octapods to each ligand: 300 μl of the desired ligand were added to 300 μl of a suspension of octapods with a concentration of 20 nM and the mixtures were kept under stirring overnight at 80 °C. HPA, ODPA, TOPO, and ODT were prepared in toluene at 0.1 mM. The resulting suspensions of octapods were washed twice via precipitation with methanol and centrifugation and finally re-dispersed in hexane for the assembly experiments.

**Interfacial self-assembly**. The air–liquid interfacial self-assembly was realized as described in literature,[27] by using Teflon wells with a volume capacity of 4.5 ml that was filled with 2 ml of diethylene glycol. In total 20 μl of the sonicated suspensions of NCs in hexane were drop-cast on the top of the diethylene glycol and the wells were closed for 10 min. After that, the wells were opened to allow complete eva-poration of the hexane layer. The yellow-colored floating membrane formed on the diethylene glycol was then collected on the desired substrates (C-coated TEM Cu grids and P-type Si substrates of $5 \times 5$ mm$^2$). The collected membranes were dried on a hot plate at 130 °C to remove any glycol trace.

**Structural, optical, and surface characterization**. Bright-field transmission electron microscopy (TEM) analyses were conducted on a JEOL JEM 1011 microscope, operated at 100 kV acceleration voltage and equipped with a Tungsten thermionic electron source. Series of tilted HAADF-STEM images were acquired using a FEI Tecnai G2 F20 TEM, operated at 200 kV, by tilting the samples from $-74°$ to $+74°$, in steps of 2° at high angles and 5° between $-30°$ and $+30°$. Alignment of tilt series was performed by using the fiducial-less alignment routine from IMOD. Volume reconstruction was done by back projection using the Tomoj plugin of ImageJ[39]. Volume rendering was performed using the software UCSF Chimera[40]. High-resolution scanning electron microscopy analyses were con-ducted on a FEI Nova 600 NanoLab instrument.

XRD analysis was carried out on a Rigaku SmartLab X-ray powder diffractometer equipped with a 9 kW CuKα rotating anode, operating at 40 kV and 150 mA. A Göbel mirror was used to both convert the divergent X-ray beam into a parallel beam and to suppress the Cu Kβ radiation. Samples were prepared by drop casting the octapod suspensions on a zero-diffraction Si substrate. Further analysis of the collected XRD patterns was conducted to estimate the octapod size by using a non-linear least square function fitting of the spectra. We followed the Whole Powder Pattern Decomposition (WPPD) Pawley method, combined with the Fundamental Parameters (FP) approach[41,42]. Such standard-less quantitative analysis allows for a precise fitting of the instrumental-broadened profile of the collected XRD patterns based on instrumental parameters, without the need of further analysis of reference samples. In the analysis, an ellipsoidal morphology was assumed for all the eight pods of the nanocrystals. Volume-weighted log-normal size distributions of the octapods, and their average pod lengths and diameters were obtained through the PDXL 2.7.2.0 Rigaku software for advanced data analysis. Relative standard deviations (RSD) below 0.25 were found for the estimated octapod average dimensions ($PI$ and $D$), except from octapods with an $L/D$ of 6.0 (RSD of 0.4), that is attributed to the presence of by-products in the form of nanorods, when decreasing the seed content, as it is noted in the Supplementary Note 2.

The absorption spectra of octapods were collected from chloroform suspensions by using a Varian Cary 5000 ultraviolet–visible–near infrared (UV–vis–NIR) spectrophotometer and the photoluminescence spectra were recorded with a Horiba FluoroMax 4 spectrometer, exciting at 340 nm and filtering the emitted light with a low-pass filter at 370 nm.

The surface of both as-synthesized and ligand-exchanged octapods was characterized using a Fourier transform infrared spectrometer (Equinox 70 FT-IR, Bruker) coupled to an attenuated total reflectance (ATR) accessory (MIRacle ATR, PIKE Technologies). The measurements were conducted on dried samples after drop casting an aliquot of 2 μl of each octapod suspension in toluene on the surface of the ATR crystal. The analysis was conducted in the operating range from 4000 to 600 cm$^{-1}$ with a resolution of 4 cm$^{-1}$ and 128 scans averaged for each spectrum. For the acquisition of the $^{31}$P-NMR spectra a Bruker Avance III 400 MHz spectrometer, equipped with a Broad Band Inverse probe (BBI). Before acquisition,

the matching and tuning were optimized on both $^1$H and $^{31}$P nuclei, and the resolution automatically adjusted. After 45° pulse calculation, 40960 transients were accumulated – without steady state scans at a temperature of 300 K, over a spectral width of 396 ppm (offset at 0 ppm), at a fixed receiver gain (2050), and using 0.30 s of inter pulses delay. An exponential line-broadening (5 Hz) was applied to FIDs (Free Induction Decay) before Fourier transform. All the $^{31}$P-NMR chemical shifts were referred to 0 ppm by using triethylphosphine as a reference.

**Monte Carlo simulations of the self-assembly**. A standard isothermal-isobaric ($NPT$) Monte Carlo approach was employed to study the self-assembled structures that form for this system, where both translation and rotation moves were used. We constrained the translational motion of the octapods' centers of mass to a two-dimensional (2D) plane to model the self-assembly taking place at the liquid–liquid interface in the experiment and used $N=200$ octapods throughout. The pressure $P$ was used to compact the system. That is, we started our simulations from a dilute gas phase with typical octapod area density $\rho \ll 0.001$. We increased the pressure $P$ from zero to its desired value over $10^5$ MCS (1 Monte Carlo Sweep = 1 move per particle per sweep), next we allowed the system to equilibrate for $10^6$ MCS, before we performed a production run of $5 \times 10^6$ MCS. During this production run, we monitored the density and temperature to examine whether the structures formed during equilibration undergo transitions. We typically performed simulations for pressures well into the range where the systems form dense liquids, gels, and solids: $P>10^{-2}Nk_BT/L^3$. For a select set of systems, we varied the pressure, see Supple-mentary Figs. 22–26 for these results.

**Data availability**. All data is available from the authors upon reasonable request.

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

## Acknowledgements

L.M. acknowledges support from the European Union through the FP7 starting ERC Grant NANO-ARCH (Contract Number 240111). J.d.G. acknowledges support from a Marie Skłodowska-Curie Intra European Fellowship (G.A. No. 654916) within Horizon 2020.

## Author contributions

A.C. performed the synthesis of octapods nanocrystals and conducted the self-assembly experiments, structural characterization, and surface analysis through Fourier transform infrared spectroscopy of the particles. J.d.G. conceived, performed, and analyzed the Monte Carlo simulations. L.G. performed the high-resolution nuclear magnetic resonance analysis of the particles. S.M. conducted X-ray diffraction characterization. R.B. collected and analyzed high angle annular dark field scanning transmission electron microscopy images of octapods, and created 3D reconstructions. L.M., J.d.G., A.C., and M.A. contributed to the understanding of the ligand distribution from the surface analyses. M.A. designed the study and supervised the project. All authors discussed the results and co-wrote the manuscript.
