## [Peer Review File(PDF 466 kb) · Nature Communications]

Reviewers' comments:

Reviewer #1 (Remarks to the Author):

This paper describes the self-assembly of octapod at the liquid:air interface depending on the shape of the particles and their surface ligand. In the first part, the assembly of octapods with their native ligand (a complex mixture of phosphorous ligand) is studied as a function of their shape. It is shown that the self-assembled structure varies from square lattices to interlocked chain. To rationalize their findings, the authors conducted an IR study to identify the surface ligand covering the particles. Computer simulations are then conducted with variation of the interaction potential between the tips and shafts of the particles and compared to the experimental results. Finally, the native ligands are replaced by other ligands (thiols and an amine).

The paper reports a very important amount of experimental findings. Almost all the possible parameters are varied. This makes the paper very comprehensive but also a bit difficult to follow since one has to constantly go from the main text to the SI. It reports novel experimental results which are clearly worth publishing. However, there are two problems with the paper:

First, the work follows similar work by the same group and I have difficulties finding out what is really new here. 7 other papers have been published on this precise subject with similar phase diagrams and simulations. The novel point here seems to be the IR study of the ligand and the explanation that the variety of structures obtained is due to the arrangement of ligands on the tip vs shaft but these structures had been obtained with the same system before and explained without the complex coating model. The authors have to address this point and explain how this work is situated within the context of their previous findings.

Second, I have several questions / concerns which prevent my recommendation to publish the current version.

- the authors claim that the particles assemble into a square lattice when the aspect ratio is small. For me this is not clearly visible on figure 1a where I do not see long range order. To prove their point, the authors should provide more compelling evidence such as higher magnification TEM images. Furthermore, a square lattice should be visible in the electron diffraction or FT of the image which is not provided here.

- The interpretation of the infrared spectra has to be rewritten because it is very difficult to follow in the present form. Some parts are in the main text, others are in the SI. I recommend that the whole interpretation is moved to the main text and that a multi-panel figure is made from figure 2b, S9 and S10. The same symbols for the different aspect ratios should be used in the S9 and S10 figures to help the reader.

In the present form, the reasoning and different hypotheses are too difficult to follow to be convincing and it is hard to see if the conclusions are adequately supported by the facts. This part is very important because the ligand model of the authors solely relies on these experiments. Furthermore, this is the main novel point of the current publication compared to previous work. Publication can be granted unless it is convincing enough which is not the case yet.

- why the ratio RCH_n varies between 3.8 and 4.05 whereas CH_2 to CH_3 ratio varies from 5 to 17 ?

- I do not understand how ligand mediated interactions can be repulsive on one part of the particle and attractive on another part. Considering the scheme of figure S13, this means that interaction abruptly shifts from attractive to repulsive from the blue to the red regions of the particle. That seems

completely unphysical to me and would be happy to hear a physical explanation from the authors.

- the model is different to the one exposed by the same authors in previous publications. How this new model brings new insights and is more predictive than the previous more realistic model ?
- the statement "we only observed self-assembled structures when ϵ_s and ϵ_T are both nonzero" is contradictory with figure 3. In particular, one of the two is 0 in cases b,d and e.
- the effect of the density of octapod on the self-assembly should be presented before the simulations so that experimental results are grouped.
- why presenting in figure 4 experiments with the exchanged ligands performed at much small concentrations (1mM, 0.5 mM) than in the native ligand case ? Furthermore the concentration is changed at the same time so that it is difficult to draw any conclusion.
- some figures in SI should be put in the main manuscript, for example S24 and S28.

Other details:

- the synthesis of the octapod should be briefly explained at the beginning of the paper.
- some TEM images should be included in figure 1 as well as shots from the tomographic reconstruction.
- i am surprised that precise size distributions for the octapods can be extracted solely from the X-ray diffraction pattern. This should be detailed more than the current two sentences in the methods section.
- how the concentration of 5 nM octapods determined ? to what value of absorbance does it correspond ? The authors should be more precise on this points for others to be able to reproduce their work.

Reviewer #2 (Remarks to the Author):

Castelli et al. report the self-assembly of octapodal shaped nanocrystals into two-dimensional superlattices. The aspect ratio of individual octapods was systematically varied and the influence on the self-assembled superstructures was studied. The experimental findings were further supported by Monte Carlo simulations. Overall, the work is of high quality and the manuscript is well-written. Therefore, this reviewer supports its publication in Nature Communications once the following issues/questions are addressed:

1. The authors make some arguments about the relative population of different ligand molecules on the relative peak intensities of several signature peaks from the FTIR spectra. It is well known ATR-FTIR measurement has a finite penetration depth into the sample films, which is around 100 nm. This would correspond to at most two layers of octapods, given their size. Therefore, the relative orientations of octapods or the superlattice structures could easily bias the relative peak ratios in the FTIR spectra. It would be more convincing if the authors can use other molecular spectroscopic tools such as NMR to assess the ligand composition.

2. The authors mentioned that ODPA ligand exchange did not product stable colloidal solution but OLAM ligand exchange did. This is counterintuitive as ODPA is a stronger ligand for CdSe compared to OLAM. Some explanations about this observation would be helpful.

Reviewer #3 (Remarks to the Author):

The manuscript of Arciniegas et al reports on self-assembly of Cu₂Se octapods into highly ordered structures. The self-assembly is found to depend on octapod's geometrical parameters and the surface ligands. This is kind of expected, but the this particular system is not trivial and extends our understanding of the self-assembly at nanoscale. The technical side of the work is rather strong. This paper can be published in Nature Communication after minor revision.

1. Please, make sure that all SEM images in Figure 4 have the same scale. Also I will recommend to redo this figure and place the graphical sketches explaining the organization of octapods in the close proximity with the corresponding SEM images.
2. Why different concentrations are used in obtained structures presented in Figure 4.
3. I will recommend to blend the experiment with theory. So far it looks like you have two separate papers.
4. Please, mention the composition of your octapods earlier in the text, as well as in the abstract.
5. Please, make your concluding statements in the abstract more defined. In my opinion, the abstract is the weakest part of this manuscript and it should be re-written.

Response to the reviewers' comments.

Reviewer #1:

We thank the reviewer for their careful reading of our manuscript and, in principle, supporting its publication in *Nature Communications*, after addressing major points. We have considered the reviewer's comments and have made a substantial effort to improve the readability and quality of our manuscript.

Remarks to the Author:

This paper describes the self-assembly of octapod at the liquid:air interface depending on the shape of the particles and their surface ligand. In the first part, the assembly of octapods with their native ligand (a complex mixture of phosphorous ligand) is studied as a function of their shape. It shown that the self-assembled structure vary from square lattices to interlocked chain. To rationalize their findings, the authors conducted a IR study to identify the surface ligand covering the particles. Computer simulations are then conducted with variation of the interaction potential between the tips and shafts of the particles and compared to the experimental results. Finally, the native ligands are replaced by other ligands (thiols and an amine). The paper reports a very important amount of experimental findings. Almost all the possible parameters are varied. This makes the paper very comprehensive but also a bit difficult to follow since one has to constantly go from the main text to the SI. It reports novel experimental results which are clearly worth publishing. However, there are two problems with the paper:

1. First, the work follows similar work by the same group and I have difficulties finding out what is really new here. 7 other papers have been published on this precise subject with similar phase diagram and simulations. The novel point here seems to be the IR study of the ligand and the explanation that the variety of structures obtained is due to the arrangement of ligands on the tip vs shaft but these structures had been obtained with the same system before and explained without the complex coating model. The authors have to address this point and explain how this work is situated within the context of their previous findings.

Response: We thank the reviewer for pointing out that we did not clearly convey the novelty of our work in our original manuscript. We have indeed studied the formation of three- and two-dimensional (3D and 2D) ordered aggregates made of this type of nanocrystals in previous works [Ref. 2 and 38 in the revised manuscript], as well as their assembly behaviour in polymers [Ref. 34 and 36 in the revised manuscript], and in binary planar superlattices [Ref. 29 in the revised manuscript]. However, in our analysis of these results, we found that we could clarify some, but not all observations using a range of increasingly refined models. This led us to conduct a detailed analysis of the octapod surfaces, to assess how the surface affects their self-assembly behavior and complete the picture we had formed from our previous experimental efforts.

Specifically, we considered the role of surface-bound ligands that are essential to the synthesis of our octapods, based on evidence in the literature that ligand-mediated interactions can play a significant role during the self-assembly of simpler particle shapes [Ref. 25 and 30 in the revised manuscript]. Here, we have correlated for the first time this analysis with the octapods' capability to form a variety of planar superlattices, accounting for the impact of octapod geometry, ligand distribution, and octapod size. This was possible thanks to an improved synthesis protocol, which allowed us to fabricate monodispersed particles with different pod lengths, including octapods with very short pods (small aspect ratio L/D of 3.0), that we were not able to produce before; as mentioned in the introduction (page 3). By selecting an interfacial self-assembly technique, we are reporting here larger domains of planar superlattices containing a single-type configuration, when compared with the solvent-mediated drop-casting approach that was used in a previous work of our group [Ref. 38 in the revised manuscript], in that case for octapods with $L/D > 4.0$.

These experimental results allowed us to further formulate a simple theoretical model that captures all observed self-assemblies and sheds new light on previous results. Importantly, our new model, that accounts for the cone-like potential interaction, which we inferred from our experiments on ligand distribution, predicts the assembly behaviour of octapods. This new model does not require an extra repulsion force of unknown origin, as we had to include in our previous works [Ref. 2 and 35 in the revised version], to explain the interlocking behaviour observed from high aspect ratio octapods. We have added sentences to further clarify the novelty of our work within the context of previous finding (“To date, ... unexplored.”; page 3).

2. Second, I have several questions/concerns which prevents my recommendation to publish the current version:

2.1. The authors claim that the particles assemble into a square lattice when the aspect ratio is small. For me this is not clearly visible on figure 1a where i do not see long range order. To prove their point, the authors should provide more compelling evidence such as higher magnification TEM images. Furthermore, a square lattice should be visible in the electron diffraction or FT of the image which is not provided here.

Response: The referee is correct in asserting that the ordering obtained from octapods with a small aspect ratio has a short range, as we mentioned in the description of the assembly on page 6. The observed planar superlattices are composed of a mosaic of domains where octapods assume different orientations, as it is also captured by our theoretical analysis (See Fig. S17 in the revised supplementary information, for $\varepsilon_S = -1.0$, and ε_T in the range from -5.0 to 5.0). Following the reviewer’s suggestions, we have collected large area HAADF-STEM images of the superlattices and performed Fast Fourier Transform (FFT). We found that the octapod-octapod interdistance remains constant in all the analyzed domains, with an average value of ca. 25.4 nm. We have also acquired high magnification HAADF-STEM images of some of the domains and conducted FFT analysis. The results evidence that there are small domains that exhibit a perfect square-lattice configuration, (i.e. 90° angles between spots and equal distances), while octapods adopt different orientations with respect to their closer particles in other regions. These analyses are now included in the Supplementary Figs. S10 and S11. We have also added sentences to provide a better description of the planar superlattices formed by these octapods (“... forming ... Fig. S11.”; page 7).

2.2. The interpretation of the infrared spectra have to be rewritten because it is very difficult to follow in the present form. Some part are in the main text, others are in the SI. I recommend that the whole interpretation is moved to the main text and that the a multi-panel figure is made from figure 2b, S9 and S10. The same symbols for the different aspect ratio should be used in the S9 and S10 figures to help the reader. In the present form, the reasoning and different hypotheses are too difficult to follow to be convincing and it is hard to see if the conclusions are adequately supported by the facts. This part is very important because the ligand model of the authors solely relies on these experiments. Furthermore, this is the main novel point of the current publication compared to previous work. Publication can be granted unless it is convincing enough which is not the case yet.

Response: We thank the reviewer for pointing out that our interpretation of the spectra was difficult to follow. We have now included a complete FTIR discussion in the main document (“To elucidate ... is at $L/D \approx 5$.”; pages 7-10), and we have also added a multi-panel figure dedicated to the FTIR analysis (new Fig. 3), which includes Fig. S9 and S10 from the previous supplementary information. We have also modified our description of the R_{P_n/CH_2} , R_{P_n/CH_3} , and R_{CH_n} ratios between the deconvoluted areas of the mentioned peaks from the FTIR spectra, and named them as P_n/CH_2 , P_n/CH_3 and CH_2/CH_3 in the revised manuscript, in order to clarify that they provide different information about the ligands bound on the octapod’s surfaces.

2.3. Why the ratio R_{CH_n} varies between 3.8 and 4.05 whereas CH₂ to CH₃ ratio varies from 5 to 17?

Response: We thank the reviewer for pointing out this source of potential confusion. The values shown in Table S2 for CH₂/CH₃ correspond to the number of methylene functional group present in the pure ligand molecule per each methyl group, while the ratio R_{CH_n} that we have established in Fig. S9 gives the ratio in absorption peaks intensities of the asymmetric stretching vibrational modes of such functional groups located at the octapod surfaces. The R_{CH_n} values depends on the absorption of light of each functional group, and therefore, they are not a quantification of the number of CH₂ per each CH₃ group. Thus, they cannot be directly compared, as they describe different aspects of the ligand analysis. To clarify this point, we have now named the R_{CH_n} ratio as CH₂/CH₃ in the new Fig. 3b and included a better description of this in the main text of the revised manuscript (“We assess ... see Fig. 3b.”; page 9). We have also modified the name of the values reported in Table S2 and S3 for $(CH_2/CH_3)_{FG}$ and included a description in the caption, to avoid further confusion.

2.4. I do not understand how ligand mediated interactions can be repulsive on one part of the particle and attractive on another part. Considering the scheme of figure S13, this means that interaction abruptly shifts from attractive to repulsive from the blue to the red regions of the particle. That seems completely unphysical to me and would be happy to hear a physical explanation from the authors.

Response: We have clarified our reasoning in the revised manuscript (“Here, ... surface interactions.”; page 11). The key point is that there are ligand-mediated interactions and other interactions, such as electrostatics and van der Waals forces. We only account for net attraction/repulsion via the effective potentials, which is due to the combination of these (possibly competing) interactions. For example, bare pieces of octapod can attract via van der Waals (vdW) attractions, but those same patches repel when coated with sufficient ligands to overcome the vdW attraction via steric depletion. We have introduced this effective interaction model, since we do not know the exact balance between the interactions present in our system. However, we did not mean to state that ligand interactions on their own can be both repulsive and attractive.

2.5. The model is different to the one exposed by the same authors in previous publications. How this new model brings new insights and is more predictive than the previous more realistic model?

Response: This model is indeed different from our previous ones and we have made the changes and improvements clearer in the revised manuscript (page 10-11). The present model is the cumulation of several years of research and subsequent refinements of our original van-der-Waals only model, as introduced in [K. Miszta et al., Nat. Mater. 10, 872 (2011), Ref. 2 in revised manuscript]. The refined model is based on the discovery of new superstructures and improved insight in the octapods’ self-assembly, specifically the role of the ligand shell and distribution identified in our submitted manuscript. Key features present in this model that were not captured by our previous work are: the ballerina configuration, without resorting to interfacial adsorption arguments, and simple interlocking configurations, without the addition of long-ranged core-core repulsions of unknown origin. This taken together with the fact that the model is relatively simple, and can be used in a predictive fashion, suggests that it is more flexible and therefore valuable.

2.6. The statement "we only observed self-assembled structures when ϵ_s and ϵ_T are both nonzero" is contradictory with figure 3. In particular, one of the two is 0 in cases b,d and e.

Response: We thank the referee for pointing out this confusing phrase. We meant to state that we only observe self-assembly when one of the two interaction potentials is non-zero and no self-assembly when both are zero. The text has been modified to reflect this in the revised manuscript (“We only... configuration“; page 11).

2.7. The effect of the density of octapod on the self-assembly should be presented before the simulations so that experimental results are grouped.

Response: We have moved the observations related to the impact of octapod density on their self-assembly before the theoretical analysis in the revised manuscript (“Finally, ... octapods”; page 7).

2.8. Why presenting in figure 4 experiments with the exchanged ligands performed at much small concentrations (1mM, 0.5 mM) than in the native ligand case? Furthermore the concentration is changed at the same time so that it is difficult to draw any conclusion.

2.9. Some figures in SI should be put in the main manuscript, for example S24 and S28.

Response to 2.8 and 2.9: We have decided to address points 2.8 and 2.9 in one, as we believe they touch upon a common theme in the reviewer’s assessment of our work. Namely, that we have selected only a few representative images to streamline the discussion in the main manuscript, but that this has gone at the expense of clarity and overview. While it is desirable to show more information in the main text, we do not wish to flood the audience with a deluge of experimental images, hence our choice in the original Fig. 4 for images most representative of the structural change, albeit taken at a lower concentration.

The reviewer’s comments have made us reassess our selection and we have rearranged the panels in Fig. 4 (now Fig. 5 in the revised manuscript) as well as added to it parts from Figs. S24 and S28 to give a more complete overview of our work, and to allow the reader to reach our intended conclusions more readily. We have now included in the figure the images obtained from the superlattices made with 5 nM concentration for ligand-exchanged octapods and we have modified the text in the revised manuscript to further clarify this point. We have preserved the structure of our original Figs. S24 and S28 in the supplement (now Figs. S12 and S30, respectively, in the revised supplementary information), at the expense of some duplication of panels, to maintain a clear presentation.

Other details:

- The synthesis of the octapod should be briefly explained at the beginning of the paper.

Response: We have added a brief explanation of the octapod synthesis to the Synthesis section in the revised document (“Briefly, ... synthesis.”; page 4). The range in which the seed concentration was changed in the synthesis has also been added to the previous sentence.

- Some TEM images should be included in figure 1 as well as shots from the tomographic reconstruction.

Response: We have changed the panels b-e in Fig. 1 for TEM images at high magnification of single octapods for the different aspect ratios in the revised manuscript. In order to better evidence that octapods present four flat- and four sharp-terminated pods, independently of their L/D aspect ratio, we have also included in the figure a panel containing the HAADF-STEM inverted images from Fig. S5c, that were used for the tomographic volume reconstructions shown in the supplementary movies S1 and S2.

- I am surprised that precise size distributions for the octapods can be extracted solely from the X-ray diffraction pattern. This should be detailed more than the current two sentences in the methods section.

Response: The nanocrystal volume-weighted size distributions and average D and P_1 values presented in Fig. 1 were determined by using an advanced data analysis of the spectra via the Whole Powder Pattern

Decomposition (WPPD) Pawley method, combined with the Fundamental Parameters (FP) approach. This allows a precise (non-linear least square) fitting of the instrumental-broadened profile of the collected XRD patterns, starting exclusively from instrumental parameters and without the need of analysis of reference samples. The analysis was conducted with the help of the PDXL v. 2.7.2.0 Rigaku software, by assuming a rod ellipsoidal geometry. We have now added these details to the Method section in the revised manuscript (“Samples ... section S2.”; page 19), including Ref. [41] and [42]. We have also improved the description provided in the caption of figure 1 to further clarify the obtained results.

- How the concentration of 5 nM octapods determined? to what value of absorbance does it correspond? The authors should be more precise on this points for others to be able to reproduce their work.

Response: The concentration of octapods was experimentally evaluated via inductively coupled plasma optical emission spectrometry, and we used the determined concentration of Cd in ppm for the calculation of the octapod concentration in nM. We have now included full details on this important point in the Method section. We have also evaluated the absorbance for a broad range of octapod concentration, considering two different particle aspect ratios (L/D of 3.0 and 5.0), exciting at 450 nm (after the CdS rod band gap). This material has been added to the revised Supplementary Information (Fig. S7). We have also added to the Method section the description of the optical characterization techniques implemented in our work.

Reviewer #2:

We are pleased that the reviewer assesses our work as “well-written” and “of high quality”. We thank the reviewer for supporting publication of our work in *Nature Communications*, after addressing the issues raised in their thorough review. Please find below a point-by-point breakdown of the changes made to the manuscript in order to address the reviewer’s criticism.

Remarks to the Author:

Castelli et al. report the self-assembly of octapodal shaped nanocrystals into two-dimensional superlattices. The aspect ratio of individual octapods was systematically varied and the influence on the self-assembled superstructures was studied. The experimental findings were further supported by Monte Carlo simulations. Overall, the work is of high quality and the manuscript is well-written. Therefore, this reviewer supports its publication in *Nature Communications* once the following issues/questions are addressed:

1. The authors make some arguments about the relative population of different ligand molecules on the relative peak intensities of several signature peaks from the FTIR spectra. It is well known ATR-FTIR measurement has a finite penetration depth into the sample films, which is around 100 nm. This would correspond to at most two layers of octapods, given their size. Therefore, the relative orientations of octapods or the superlattice structures could easily bias the relative peak ratios in the FTIR spectra. It would be more convincing if the authors can use other molecular spectroscopic tools such as NMR to assess the ligand composition.

Response: We thank the reviewer for this observation. In our ATR-FTIR measurements we used randomly oriented octapods to avoid the possible impact of preferential particle orientations on the collected signals. To this, we prepared highly concentrated suspensions of octapods and drop cast them on the ATR crystal to obtain our samples. The octapod suspensions were dried in open air to achieve fast solvent evaporation, in order to avoid the ordering that can occur for slow solvent evaporation [Refs. 2 and 21 of the revised manuscript]. In addition to this, we ensured that the samples completely covered the ATR-FTIR spot, thus ensuring that the signal comes from a large number of octapods. We have added a sentence in the main text to clarify this important point (“To avoid ... air”; page 8).

On the other hand, we have put much effort to assess the ligand composition using other molecular spectroscopic tools. Specifically, we have implemented high-resolution Nuclear Magnetic Resonance Spectroscopy (HR-NMR) of the as-synthesized octapod suspensions. However, we faced several technical issues that prevented further analysis of the ligands:

- We performed proton NMR (^1H NMR) measurements on pure ligands prepared at 50 mM to verify the selectivity of the technique. We tested different solvents to find a common one that ensured an optimal dissolution of all the ligands. With the exception of ODPa, all the ligands were soluble in both toluene- d_8 and chloroform- d (CDCl_3) and for those ligands the spectra satisfied the criteria of selectivity for at least one resonance. In THF- d_8 , all the ligands were highly soluble, but an inefficient selectivity was obtained in the ^1H NMR spectra.
- This motivated us to conduct phosphorus-31 NMR (^{31}P NMR) measurements, from which a good selectivity in THF- d_8 was achieved for all the ligands. It should be noted, however, that HPA and ODPa exhibited extremely similar resonances (ca. 33.17 ppm and 33.19 ppm, respectively; Fig. S34 of the revised supplementary information).
- We performed the analysis on octapods with the smallest and the largest aspect ratio (L/D of 3.0 and 6.0) in THF- d_8 , for a comparative analysis. While the acquired ^{31}P NMR spectra from octapods with an

L/D of 3.0 evidenced the presence of TOP-S and TOPO, most likely a low content of the other ligands prevented their detectability also due to sensitivity limitations of the technique.

- In addition to this, the poor solubility of octapods with a larger aspect ratio in suspensions prepared at 250 nM hindered their analysis. We were only able to use an extremely low concentration of nanocrystals when compared to that typically required for the technique, ca. 30 mM. In this case, the particles precipitated in few minutes. This was also the case for octapods with the smallest aspect ratio when we increased the particle concentration in our attempts to increase the detectability of the phosphorous acids.

Nevertheless, considering the relevant experimental details that these attempts provided we have included these results in Section 10 (Fig. S34) of the revised supplementary information. We have also added a description of the implemented technique in the Method section and sentences in the main text to note further the relevance of our ATR-FTIR analysis.

2. The authors mentioned that ODPA ligand exchange did not product stable colloidal solution but OLAM ligand exchange did. This is counterintuitive as ODPA is a stronger ligand for CdSe compared to OLAM. Some explanations about this observation would be helpful.

Response: We would like to clarify this point to the referee, as there appears to be a misunderstanding on ligand affinity of ODPA versus OLAM for CdSe nanocrystals.

ODPA – in common with all phosphonic, but also carboxylic acids – is a good ligand especially when deprotonated. Phosphonic acids can lose two protons, but even by losing only one proton, they become electron-rich enough to complex Cd^{2+} ions on the surface of CdSe nanocrystals. In practice, surface Cd ions, or Cd^{2+} ions in solution, are passivated by molecules having functional groups rich in electrons. In this respect, OLAM is a better passivating/complexing agent than a protonated phosphonic acid. However, a deprotonated phosphonic acid is a better passivating agent than OLAM.

In our synthesis of CdSe nanocrystals, phosphonic acids react with CdO (also used in the synthesis): they lose part of their protons by reacting with CdO, freeing water and forming Cd-phosphonate complexes. Since in our ligand exchange procedure, we only add ODPA (there is no base to deprotonate the phosphonic acids), it will not interact so strongly with the surface of the NCs.

In addition to this, the observed limited solubility of ODPA in toluene at room temperature, even at the low concentration prepared for the ligand-exchange procedure described in the Method section, made it more difficult to produce stable suspensions of ODPA ligand-exchanged octapods. To overcome this issue, we have investigated other solvents and we found that ODPA is highly soluble in tetrahydrofuran. However, the miscibility of this solvent with the diethyl glycol sub-phase made it impossible to use this solvent for the formation of planar superlattices. We have added sentences in the main text to explain this (“The observed ... stabilization”; page 14).

Reviewer #3:

We thank the reviewer for supporting publication of our work in *Nature Communications*, after minor revision and assessing the technical aspect of our work as “strong”. Please find below our response to the criticisms made.

Remarks to the Author:

The manuscript of Arciniegas et al reports on self-assembly of Cu₂Se octapods into highly ordered structures. The self-assembly is found to depend on octapod’s geometrical parameters and the surface ligands. This is kind of expected, but the this particular system is not trivial and extends our understanding of the self-assembly at nanoscale. The technical side of the work is rather strong. This paper can be published in Nature Communication after minor revision.

1. Please, make sure that all SEM images in Figure 4 have the same scale. Also I will recommend to redo this figure and place the graphical sketches explaining the organization of octapods in the close proximity with the corresponding SEM images.

Response: We have extensively modified the images in the revised manuscript, taking into account the referees suggestions.

2. Why different concentrations are used in obtained structures presented in Figure 4.

Response: Our original manuscript contained as few images as possible, which were nonetheless representative of the main features uncovered by our experiments, in order to streamline the narrative. We acknowledge that using pictures obtained at a different concentration may have led to confusion and we have revised the manuscript accordingly.

3. I will recommend to blend the experiment with theory. So far it looks like you have two separate papers.

Response: We thank the reviewer for this suggestion. However, we have decided to maintain the structure of the original document. Our reasoning is as follows. In the current format, the theory is a natural consequence of our thorough experimental analysis. A bonus feature is that the theory makes predictions that can, and were, subsequently tested and shown to hold. A rewrite that would blend the theory and experiment more, would not bring this last, crucial point across quite as clearly. To ensure that the revised manuscript feels more like a single piece, we have smoothed the boundaries between the theory and experimental part.

4. Please, mention the composition of your octapods earlier in the text, as well as in the abstract.

Response: We have mentioned the chemical composition of the octapods in both the abstract and earlier in the main text, page 3 of the revised manuscript.

5. Please, make your concluding statements in the abstract more defined. In my opinion, the abstract is the weakest part of this manuscript and it should be re-written.

Response: We thank the reviewer for pointing out that our message was not easily distilled from the abstract and we have made a substantial effort to rewrite it more clearly and make it appealing to its intended audience.